# Host–Guest Cocrystallization of Phenanthrene[2]arene Macrocycles Facilitating Structure Determination of Liquid Organic Molecules

**DOI:** 10.3390/molecules29112523

**Published:** 2024-05-27

**Authors:** Guangchuan Ou, Yanfeng Zhang, Qiong Wang, Yingzhi Tan, Qiang Zhou, Fei Zeng

**Affiliations:** 1College of Chemistry and Bioengineering, Hunan University of Science and Engineering, Yongzhou 425199, China; wqiong1975@huse.cn (Q.W.); tanyz@huse.cn (Y.T.); zhouq7712@huse.cn (Q.Z.); 2Agricultural Comprehensive Service Center, Yongzhou 425000, China; yfzhang@huse.cn

**Keywords:** phenanthrene[2]arene, macrocyclic host, guest liquid molecules, structure determination, cocrystallization

## Abstract

Single-crystal X-ray diffraction analysis has emerged as the most reliable method for determining the structures of organic molecules. However, numerous analytes, such as liquid organic molecules, pose challenges in crystallization, making their structures directly elusive via X-ray crystallography methods. Herein, we introduced the rapid cocrystallization of a macrocycle named phenanthrene[2]arene (PTA, host) with 15 liquid organic molecules (guests). The guest liquid organic molecules were successively cocrystallized with the aid of the PTA host. Moreover, the chemical structures of the liquid organic molecules could be determined through single-crystal X-ray diffraction analysis. PTA exhibited high adaptivity and was capable of encapsulating liquid organic molecules without forming covalent bonds or strong directional interactions. The results revealed that the adaptive crystals of PTA exhibited excellent cocrystallization capacity. Weak noncovalent interactions between the host and guest molecules were crucial for organizing the guests in an ordered pattern.

## 1. Introduction

A novel organic compound, once successfully synthesized or isolated, is typically characterized via various analytical methods, such as nuclear magnetic resonance spectroscopy, mass spectrometry, and infrared spectroscopy. These methods provide valuable insights into the molecular structure of the compound. Among these methods, X-ray single-crystal diffraction has emerged as the most reliable technique for determining molecular structures. This method enables the direct observation of the spatial arrangements of atoms within molecules. However, X-ray crystallography requires a high-quality single crystal for effective analysis. Numerous organic compounds, particularly liquids or amorphous solids, pose challenges for independent crystallization. Recently, several innovative strategies have been developed to improve the crystallization process [1,2,3]. Fujita et al. [4,5,6,7,8,9] introduced the “crystalline sponge (CS)” method for determining the structures of non-crystalline compounds via X-ray single-crystal diffraction. The crystal sponge mainly used in this research comprised metal–organic frameworks [4,10,11,12] and organic molecular frameworks [13,14]. However, the crystal sponge materials may not be universally suitable for other molecules with significantly different sizes and shapes owing to geometric restrictions within the cavities. To address these limitations, several methods are currently being explored. Recently, a particularly significant method involving “crystallization chaperones” has emerged. These crystallization chaperones, including tetraaryladamantanes [15,16,17,18,19,20,21,22] and trimesic acid [23,24,25,26] readily form crystalline inclusion complexes with various guest molecules and provide accurate structural information on difficult-to-crystallize small molecules. Additionally, supramolecular assemblies, such as cyclodextrins [27] and pillararenes, ref. [28] can form crystalline inclusion complexes with simple organic molecules. Recently, Li et al. [29] used Ag_3_Pz_3_ (Pz = 3,5-bis(trifluoromethyl)pyrazole) directly to elucidate the structures of organic molecules, even in mixtures or crude extracts of natural medicine. Wang et al. [30] used cyclotrixylohydroquinoylene derivative, characterized by a negative electrostatic potential on its surface and an electron-rich cavity, to determine the various structures of organic molecules or the absolute configuration of chiral molecules through a host–guest system. As part of our research interests in supramolecular chemistry [31,32,33], this study introduced macrocyclic phenanthrene[2]arene (PTA) with giant and electron-rich cavities as a crystalline host for elucidating molecular structures (Figure 1). Fifteen liquid organic molecules were successfully cocrystallized with the PTA host, and their precise molecular structures were determined through X-ray diffraction analysis.

## 2. Results and Discussion

In our previous work, we initially designed and synthesized a PTA host molecule featuring 12 methoxyl groups arranged as follows: four methoxyl groups above the ring, four methoxyl groups in the ring plane, and the remaining four methoxyl groups below the ring. Additionally, the PTA molecule included four phenyl groups and two phenanthrene units oriented in the same plane. This configuration formed an electron-rich cavity with dimensions of 10.11 × 10.17 Å (as determined by the C···C distances) (Figure 1). The adaptive crystals of PTA exhibited excellent benzene adsorption capacity, with each host molecule capable of adsorbing up to five benzene molecules at maximum. Moreover, activated crystals of PTA can separate benzene from an equimolar mixture of benzene and cyclohexane [31]. These observations prompted the exploration of potential applications of PTA in cocrystallization and for determining the structures of liquid guest molecules via X-ray crystallography. Therefore, we selected structurally similar liquid organic guests for cocrystallization experiments with the PTA host. In the absence of any solvent, a mixture of the organic guests and the PTA host was heated until a clear solution formed. The resulting solution was then allowed to cool to room temperature. Colorless crystals suitable for X-ray diffraction were obtained after overnight crystallization. The structures of the cocrystals, containing the liquid organic guests, were confirmed via X-ray crystallography. With 15 cocrystallization structures, monoclinic crystal systems and a host/guest ratio of 1:1 for 2, 1:2 for 1, 3, and 5 to 15, and 1:3 for 4 were identified (Table 1 and Figure 2). In the asymmetric unit of the complexes, no covalent bonds or strong mutual interactions, such as hydrogen bonding, were observed between host and guest molecules. The liquid organic guest molecules solidified and crystallized with the aid of the PTA host. However, some of the liquid guest molecules exhibited a disorder due to the absence of mutual interactions with the host during cocrystallization.

### Description of Structures

PTA underwent cocrystallization with a series of liquid organic samples containing different functional groups. The crystal structures (Figure 3, Figure 4 and Appendix A) illustrated that PTA could adapt to cocrystallization with different functional groups. The crystal data yielded satisfactory *R*_1_ values after the refinement process (Appendix A). The macrocycles of complex **1** were interconnected through noncovalent interactions to form rhombic-shaped channels along the *b*-axis. These channels were occupied by guest molecules positioned appropriately along the *b*-axis. Additionally, the guest molecules were positioned both above and below the macrocyclic cavity, rather than inside the cavity along the *a*-axis (Figure 3).

Moreover, CH···π intermolecular interactions were observed between the plane of 1-chloro-3-methylbenzene and the phenanthrene unit of the PTA host, with a distance of 2.721 Å. Furthermore, CH···C intermolecular interactions were observed between the 1-chloro-3-methylbenzene and 1,4-dimethoxybenzene units of PTA, with a distance of 2.764 Å (measured by the C–H···C distances) in complex **1** (Figure 5). These weak, multiple noncovalent interactions may effectively hinder the thermal motion of guest molecules and contribute to stabilizing the crystal structure.

In all but one of the complex **4**, the guest–host complex lies on an inversion center in the crystal. Complexes **1**, **3**, and **5** to **15** crystallized in the monoclinic crystal system, except for complexes **5** and **7**, which crystallized in the triclinic crystal system with a *P*-1 space group. Each complex mentioned above contained one PTA and two guest molecules in the per-asymmetric unit. Complex **2** crystallized in the *P*2_1_/*n* space group and contained one PTA and one guest molecule. Complex **4** crystallized in the *C*2/*c* space group and contained one PTA and three guest crystallographically independent molecules (Table 1). The adjacent PTA molecules of complex **1** were arranged at 75.48° along the *a*-axis. The angles between the adjacent PTA ring planes were measured to be 0.00° to 75.65°. These planes were formed by the four carbon atoms attached to the methoxyl of phenanthrene (Figure 6). Complexes **5** and **7**, which displayed 0°, cocrystallized in the triclinic crystal system with a *P*-1 space group. Complexes **4** and **12**, which exhibited smaller angles at 33.39° and 22.99°, cocrystallized in the monoclinic crystal system with *C*2/*c* and *P*2_1_/*c* space groups, respectively. Therefore, the spatial arrangement of PTA molecules may be independent of the size of guest molecules.

X-ray single-crystal diffraction analysis enabled the direct observation of absolute configuration. Therefore, we further explored the determination of the absolute configuration of chiral molecules using PTA. However, this remained a challenging task owing to poor data quality and guest molecular disorders. Two chiral guests, (*R*)-(1-chloroethyl)benzene (GM-**5**) and (*S*)-(1-chloroethyl)benzene (GM-**6**), were investigated (Figure 2 and Table 1). Moreover, complexes **5** and **6** crystallized in the *P*-1 and *P*2_1_/*n* centrosymmetric space groups, respectively. In each crystal structure, two guest molecules related by an inversion center in the asymmetric unit exhibited opposite chirality. This phenomenon may be attributed to chiral racemization caused by heat dissolution during the experiment.

## 3. Materials and Methods

PTA was prepared according to a previously reported procedure [31]. All chemicals used in this study were commercially available and used without further purification.

### 3.1. General Methods for Cocrystallization

Initially, ~2 mg of solid PTA was placed in a cylindrical sample bottle (1 mL), along with ~40 mg of liquid analyte (equivalent to 2 drops). The resulting suspension was heated on a hot plate until a clear solution formed. Subsequently, the solution was allowed to cool to room temperature. After 24 h, colorless prism- or block-shaped crystals were obtained and analyzed via X-ray single-crystal diffraction.

### 3.2. Crystal Structure Determination

Single-crystal data were collected using a diffractometer (SuperNova Dual AtlasS2, Agilent, UK), with Cu-Kα radiation (λ = 1.54184 Å), or an Apex II diffractometer (Bruker Smart, Bruker, Germany) with Mo-Kα radiation (λ = 0.71073 Å). Empirical absorption corrections were applied using the SADABS program [34]. All structures were determined through direct methods, which yielded the positions of all non-hydrogen atoms. The positions of these atoms were initially refined isotropically, followed by anisotropic refinement. All hydrogen atoms of the ligands were placed in calculated positions with fixed isotropic thermal parameters and included in the structure factor calculations during the final stage of full-matrix least-square refinement. All calculations were conducted using the SHELXTL 5.1 software package [35]. The disorders mainly originated from the guest molecules and methoxyl of PTA. To address the disorder, some restraints and constraints were applied to refine the crystal structures. In complexes **2**, **4**, **6**, **7**, **8**, **11**, and **15**, the guest molecules were refined reasonably using a positional disorder model, ensuring that the chemical assignment remained unaffected. The graphics of crystal structures were generated using software programs such as Diamond 4.6, Mercury 3.0, and ChemBioDraw 12.0. Detailed crystallographic data are summarized in Appendix A.

## 4. Conclusions

This study introduced a highly adaptive crystalline material for the structural elucidation of liquid organic molecules that faced challenges in independent crystallization. Fifteen liquid organic molecules with different shapes and sizes were successfully cocrystallized with PTA. Crystal structure analysis revealed that the weak noncovalent interactions between the host and guest molecules played vital roles in the overall stabilization of crystal packing. The excellent capacity of PTA to cocrystallize with guest molecules enabled its use as a “crystalline sponge” material for determining the structures of liquid organic molecules.

## Data Availability

Crystallographic data for compounds **1–15** have been deposited with the Cambridge Crystallographic Data Center as supplemental publication numbers CCDC 2350448 (for **1**), 2350449 (for **2**), 2350262 (for **3**), 2350451 (for **4**), 2350450 (for **5**), 2350452 (for **6**), 2350266 (for **7**), 2350454 (for **8**), 2350455 (for **9**), 2350459 (for **10**), 2350456 (for **11**), 2350458 (for **12**), 2350457 (for **13**), 2350267 (for **14**), and 2350268 (for **15**), respectively. Copies of the data can be obtained free of charge via http://www.ccdc.cam.ac.uk, accessed on 20 May 2024.

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
