# Peer review of "Host–Guest Cocrystallization of Phenanthrene[2]arene Macrocycles Facilitating Structure Determination of Liquid Organic Molecules"

_molecules, 2024, doi:10.3390/molecules29112523_

Round 1

Reviewer 1 Report

Comments and Suggestions for Authors

The manuscript reports use of guest-host complexes to determine the “precise molecular structures” of guests that are liquids at room temperature. The authors succeeded in getting the structures of 15 such complexes, although except for the two that were determined at low temperature (1 and 2), the precision of the determinations is rather low, to a large degree because of disorder in most of the structures. They were also unsuccessful in getting structures of chiral molecules, probably as a result of racemization of the guest brought on by heating the sample in their preparative method.

In all but one of the structures (4), the guest-host complex lies on an inversion center in the crystal, an important detail not mentioned anywhere in the manuscript. Instead, the authors make much of the space groups, but frequently do so incorrectly. For instance, line 85 seems to say that all structures are monoclinic, but 5 and 7 are triclinic. Lines 135-138 and 156 give incorrect space groups for nearly all the compounds, disagreeing with Table 1, in which they are all correct. Also, the authors should note that P21/n and P21/c are the same space group, with the axes labeled differently. So, it would be better to use terms such as “the P21/n setting” if they want to distinguish between the two. But, as I mention above, it is more important to give the symmetry of the site on which the complex lies rather than spend lines in the text giving the space groups.

Concerning 5 and 6, for which a homochiral guest molecule racemizes to give centrosymmetric crystals, it is said that “two independent guest molecules in the asymmetric unit exhibited opposite chirality.” However, the two are not independent, but related by an inversion center. It would be interesting to see if crystalline chiral guest-host complexes could be prepared without heating, but sitting at room temperature for an extended period of time. I assume that the chiral guests are stable at room temperature. Structures 5 and 6 differ in that 6 has not only the two guest mirror-image organics but also apparently a water molecule, which is not mentioned in the text. The water H atoms are in impossible positions, and I’m sure they were difficult to locate in the middle of all the disorder. It would be better to just leave them out of the refinement, but it should be made clear in the manuscript that there is solvent present in this structure.

In compound 12, the guest molecule in the asymmetric unit is not in contact with the host, but molecules related to it by 1-x, ½+y, ½-z and also x, ½-y, ½+z are. It would be better to transform the guest molecule to one of these positions and redo the refinement.

While some structures have atoms numbered in reasonable fashion, structures 3,4,10 and 14 do not. These have random atom numbers, apparently assigned by the structure-solution software.

This makes it difficult for readers (and reviewers!), and it should take only a few minutes to renumber the atoms and do a few more cycles of refinement. This should be done for these four structures.

Comments on the Quality of English Language

In comments

Author Response

Review 1

Comments and Suggestions for Authors:

The manuscript reports use of guest-host complexes to determine the “precise molecular structures” of guests that are liquids at room temperature. The authors succeeded in getting the structures of 15 such complexes, although except for the two that were determined at low temperature (1 and 2), the precision of the determinations is rather low, to a large degree because of disorder in most of the structures. They were also unsuccessful in getting structures of chiral molecules, probably as a result of racemization of the guest brought on by heating the sample in their preparative method.

In all but one of the structures (4), the guest-host complex lies on an inversion center in the crystal, an important detail not mentioned anywhere in the manuscript. Instead, the authors make much of the space groups, but frequently do so incorrectly. For instance, line 85 seems to say that all structures are monoclinic, but 5 and 7 are triclinic. Lines 135-138 and 156 give incorrect space groups for nearly all the compounds, disagreeing with Table 1, in which they are all correct. Also, the authors should note that P21/n and P21/c are the same space group, with the axes labeled differently. So, it would be better to use terms such as “the P21/n setting” if they want to distinguish between the two. But, as I mention above, it is more important to give the symmetry of the site on which the complex lies rather than spend lines in the text giving the space groups.

Author Response:“In all but one of the complex 4, the guest-host complex lies on an inversion center in the crystal” was added in line 139. The description of space groups has been revised in lines 140-142 and 161. “Additionally, complex 12 crystallized in the P21/n space group, while the remaining complexes also crystallized in the P21/n space group.” was deleted in line 142.

Concerning 5 and 6, for which a homochiral guest molecule racemizes to give centrosymmetric crystals, it is said that “two independent guest molecules in the asymmetric unit exhibited opposite chirality.” However, the two are not independent, but related by an inversion center. It would be interesting to see if crystalline chiral guest-host complexes could be prepared without heating, but sitting at room temperature for an extended period of time. I assume that the chiral guests are stable at room temperature. Structures 5 and 6 differ in that 6 has not only the two guest mirror-image organics but also apparently a water molecule, which is not mentioned in the text. The water H atoms are in impossible positions, and I’m sure they were difficult to locate in the middle of all the disorder. It would be better to just leave them out of the refinement, but it should be made clear in the manuscript that there is solvent present in this structure.

Author Response:“In each crystal structure, two independent guest molecules in the asymmetric unit exhibited opposite chirality” was updated to “In each crystal structure, two guest molecules related by an inversion center in the asymmetric unit exhibited opposite chirality” in line 162.

The host do not dissolve well in guest molecules without heating.

“Each complex contained one PTA and two guest crystallographically independent molecules, in the per asymmetric unit” was updated to “Each complex mentioned above contained one PTA and two guest molecules in the per asymmetric unit” in line 142-143.

 The water molecule in structure 6, erroneous assignment of atom type, has been well dealt with.

In compound 12, the guest molecule in the asymmetric unit is not in contact with the host, but molecules related to it by 1-x, ½+y, ½-z and also x, ½-y, ½+z are. It would be better to transform the guest molecule to one of these positions and redo the refinement.

 Author Response:The refinement has been done in compound 12.

While some structures have atoms numbered in reasonable fashion, structures 3,4,10 and 14 do not. These have random atom numbers, apparently assigned by the structure-solution software.

This makes it difficult for readers (and reviewers!), and it should take only a few minutes to renumber the atoms and do a few more cycles of refinement. This should be done for these four structures.

Author Response:The atoms of structures 3,4,10 and 14 have been numbered.

Reviewer 2 Report

Comments and Suggestions for Authors

The paper “Host–Guest Cocrystallization of Phenanthrene[2]arene Macrocycles Facilitating Structure Determination of Liquid Organic Molecules by Guangchuan Ou et al. is an interesting study of a series of cocrystals comprising a macrocyclic host and various aromatic guest molecules. The paper is interesting, for the most part well written, and I would recommend it to be published in Molecules after some minor revisions.

There is however one potentially serious concern regarding the determined crystal structures 5 and 6 (derived from the two enantiomers of (1-chloroethyl)benzene). Both structures have been solved and refined in centrosymmetric space groups (P-1 and P21/n), i.e. with both enantiomers present in the crystal structure, and this is accounted for in the discussion by presuming that racemisation has occurred upon heating of the reactant mixture. However, there are some indications that these are in fact cases of pseudosymmetry, and that the structures are in fact non-centric, erroneously refined as centrosymmetric (i.e. there was no racemisation, but it is only the artefact of the wrongly assigned space group). This is not unlikely to occur if the majority of the atoms (the PTA macrocycle) form an (approximately) centrosymmetric assembly, and only a small portion of the structure (the guest molecule) does not. Classic 'symptom' of such misassignment of the space group apparent disorder of the part of the structure which 'brakes' the symmetry. The disorder in 6 is apparent, but is also present in 5, as evidenced by large and highly elongated (even with ISOR!) displacement ellipsoids of C1 and Cl1. Therefore both 5 and 6 should be solved and refined in non-centric groups (viz P1 and P21), in order to see whether these problems can be resolved. It might be necessary to perform additional measurements (in particularly for 5, as the whole reflection sphere is needed for proper structure refinement in P1).

Structure 6 has an additional highly suspect feature – a water molecule (of unknown origin) positioned 1.9191 A (extremely close contact!) from a phenyl carbon (C1BA), with the 'water molecule' hydrogen atoms do not participate in any sensible HB interactions. This is highly unusual, and implies erroneous assignment of atom type (possibly the O7 is in fact a chlorine atom with ca 0.25 occupancy from a differently oriented (1-chloroethyl)benzene molecule). Additional problem for supposing this to be a water molecule is in the fact that there was no water in the system upon crystallisation, and, since the both the crystallisation medium and the binding site in the crystal structure are highly hydrophobic, absorption of water from the atmosphere is extremely unlikely. Therefore, the claim of the presence of water in this structure should be corroborated by some additional evidence, such as a TGA curve showing the loss of a stoichiometric amount of water.

Both structures therefore, in my opinion, require a more detailed inspection.

Apart from this concern, I have just a few additional suggestions/comments:

1. Have the authors considered to perform some TGA/DSC measurements? As the obtained solids are a relatively large series of closely related which comprise a constituent which is, on its own, a liquid at ambient conditions, some data on thermal stability of these compounds (and possibly correlating this with composition/structure of the cocrystal or melting/boiling point of the guest) would be most interesting.

2. The fact that 5 and 6 crystallise in different space groups is indicative of polymorphism (normally, one would expect two enantiomers to form crystals belonging to the same (or enantiomerically equivalent) space group. The fact that in one case a triclinic and in the other a monoclinic crystal was obtained, indicates that both crystal forms are possible in both cases. Furthermore, if racemisation has indeed occurred (and both 5 and 6 comprise racemic (1-chloroethyl)benzene, as neither is isostructural with 4, it would seem that racemic (1-chloroethyl)benzene forms (at least) 3 different polymorphs under the same conditions. It is therefore possible that in all three cases there were several different crystal types present. Have the powder diffraction patterns of the bulk material been measured?

3. I would suggest the Table 1 to be modified by being more precise the composition of the crystals obtained – if indeed in 5 and 6 racemisation has occurred, and 6 also comprises water molecules, simple designation of the number of ‘guest molecules’ is not sufficient information (the ‘guest’ is no longer the same substance as that named in column 2). However, see also comment above on structures of 5 and 6).

4. Page 2, lines 52-54: the abbreviations ‘Ag3Pz3’ and ‘CTX[P(O)Ph]’ have no meaning out of context – a fuller formula or name is required.

5. Page 3, lines 75-76 ‘. Solvent molecules, such as benzene and cyclohexane, tended to retain the crystal structure and can also be used to cultivate single crystals of PTA.’ – it is not entirely clear what is meant by this sentence.

6. Page 3, lines 83, 85 – ‘cocrystallization structures’ – perhaps, ‘structures of the cocrystals’?

7. Page 6, lines 135-139 – this section is rather confusingly written, and is apparently in odds with the data in the actual crystallographic data (as present in the cif-s and Table 1)

8. Page 7, line 184 ‘this order’ – probably ‘the disorder’?

Comments on the Quality of English Language

There some additional minor grammatical and syntactical errors – some additional attention to the language should be given.

Author Response

Review 2

Comments and Suggestions for Authors:

The paper “Host–Guest Cocrystallization of Phenanthrene[2]arene Macrocycles Facilitating Structure Determination of Liquid Organic Molecules by Guangchuan Ou et al. is an interesting study of a series of cocrystals comprising a macrocyclic host and various aromatic guest molecules. The paper is interesting, for the most part well written, and I would recommend it to be published in Molecules after some minor revisions.

There is however one potentially serious concern regarding the determined crystal structures 5 and 6 (derived from the two enantiomers of (1-chloroethyl)benzene). Both structures have been solved and refined in centrosymmetric space groups (P-1 and P21/n), i.e. with both enantiomers present in the crystal structure, and this is accounted for in the discussion by presuming that racemisation has occurred upon heating of the reactant mixture. However, there are some indications that these are in fact cases of pseudosymmetry, and that the structures are in fact non-centric, erroneously refined as centrosymmetric (i.e. there was no racemisation, but it is only the artefact of the wrongly assigned space group). This is not unlikely to occur if the majority of the atoms (the PTA macrocycle) form an (approximately) centrosymmetric assembly, and only a small portion of the structure (the guest molecule) does not. Classic 'symptom' of such misassignment of the space group apparent disorder of the part of the structure which 'brakes' the symmetry. The disorder in 6 is apparent, but is also present in 5, as evidenced by large and highly elongated (even with ISOR!) displacement ellipsoids of C1 and Cl1. Therefore both 5 and 6 should be solved and refined in non-centric groups (viz P1 and P21), in order to see whether these problems can be resolved. It might be necessary to perform additional measurements (in particularly for 5, as the whole reflection sphere is needed for proper structure refinement in P1).

Structure 6 has an additional highly suspect feature – a water molecule (of unknown origin) positioned 1.9191 A (extremely close contact!) from a phenyl carbon (C1BA), with the 'water molecule' hydrogen atoms do not participate in any sensible HB interactions. This is highly unusual, and implies erroneous assignment of atom type (possibly the O7 is in fact a chlorine atom with ca 0.25 occupancy from a differently oriented (1-chloroethyl)benzene molecule). Additional problem for supposing this to be a water molecule is in the fact that there was no water in the system upon crystallisation, and, since the both the crystallisation medium and the binding site in the crystal structure are highly hydrophobic, absorption of water from the atmosphere is extremely unlikely. Therefore, the claim of the presence of water in this structure should be corroborated by some additional evidence, such as a TGA curve showing the loss of a stoichiometric amount of water.

Both structures therefore, in my opinion, require a more detailed inspection.

 Author Response:

For 5, the refinement cannot convergence when solved in P1 space group. Thus received the following checkCIF A and B alerts.

PLAT080_ALERT_2_A Maximum Shift/Error ............................       1.41 Why ?

PLAT340_ALERT_3_B Low Bond Precision on  C-C Bonds ...............    0.01537 Ang.

For 6, the refinement cannot convergence when solved in P21 space group. Thus received the following checkCIF A and B alerts.

PLAT080_ALERT_2_A Maximum Shift/Error ............................       0.42 Why ?

PLAT340_ALERT_3_B Low Bond Precision on  C-C Bonds ...............     0.0153 Ang.

See attached CIF files.

The water molecule in structure 6, erroneous assignment of atom type, has been well dealt with.

Apart from this concern, I have just a few additional suggestions/comments:

  1. Have the authors considered to perform some TGA/DSC measurements? As the obtained solids are a relatively large series of closely related which comprise a constituent which is, on its own, a liquid at ambient conditions, some data on thermal stability of these compounds (and possibly correlating this with composition/structure of the cocrystal or melting/boiling point of the guest) would be most interesting.

Author Response:The TGA/DSC and data on thermal stability of these compounds were not measured due to the small sample ~40 mg. I will consider these characterization analysis in future experiments.

  1. The fact that 5 and 6 crystallise in different space groups is indicative of polymorphism (normally, one would expect two enantiomers to form crystals belonging to the same (or enantiomerically equivalent) space group. The fact that in one case a triclinic and in the other a monoclinic crystal was obtained, indicates that both crystal forms are possible in both cases. Furthermore, if racemisation has indeed occurred (and both 5 and 6 comprise racemic (1-chloroethyl)benzene, as neither is isostructural with 4, it would seem that racemic (1-chloroethyl)benzene forms (at least) 3 different polymorphs under the same conditions. It is therefore possible that in all three cases there were several different crystal types present. Have the powder diffraction patterns of the bulk material been measured?

Author Response:The powder diffraction patterns of the bulk material were not measured due to the small sample ~40 mg. The racemic (1-chloroethyl)benzene maybe forms 3 different polymorphs under the same conditions.

  1. I would suggest the Table 1 to be modified by being more precise the composition of the crystals obtained – if indeed in 5 and 6 racemisation has occurred, and 6 also comprises water molecules, simple designation of the number of ‘guest molecules’ is not sufficient information (the ‘guest’ is no longer the same substance as that named in column 2). However, see also comment above on structures of 5 and 6).

Author Response:The Table 1 has been modified.

  1. Page 2, lines 52-54: the abbreviations ‘Ag3Pz3’ and ‘CTX[P(O)Ph]’ have no meaning out of context – a fuller formula or name is required.

Author Response:The abbreviations ‘Ag3Pz3’ and ‘CTX[P(O)Ph]’ are revised.

  1. Page 3, lines 75-76 ‘. Solvent molecules, such as benzene and cyclohexane, tended to retain the crystal structure and can also be used to cultivate single crystals of PTA.’ – it is not entirely clear what is meant by this sentence.

Author Response:‘Solvent molecules, such as benzene and cyclohexane, tended to retain the crystal structure and can also be used to cultivate single crystals of PTA.’ was deleted.

  1. Page 3, lines 83, 85 – ‘cocrystallization structures’ – perhaps, ‘structures of the cocrystals’?

Author Response:‘cocrystallization structures’ updated to ‘structures of the cocrystals’.

  1. Page 6, lines 135-139 – this section is rather confusingly written, and is apparently in odds with the data in the actual crystallographic data (as present in the cif-s and Table 1)

Author Response:This section was inspected and revised.

  1. Page 7, line 184 ‘this order’ – probably ‘the disorder’?

Author Response:‘this order’ updated to ‘the disorder’.

Round 2

Reviewer 1 Report

Comments and Suggestions for Authors

Make certain that the revised CIFs are deposited with the CCDC as replacements.